# The Signal Transducer IL6ST (gp130) as a Predictive and Prognostic Biomarker in Breast Cancer

**DOI:** 10.3390/jpm11070618

**Published:** 2021-06-29

**Authors:** Carlos Martínez-Pérez, Jess Leung, Charlene Kay, James Meehan, Mark Gray, J Michael Dixon, Arran K Turnbull

**Affiliations:** 1Breast Cancer Now Edinburgh Research Team, MRC Institute of Genetics and Cancer, Western General Hospital, University of Edinburgh, Edinburgh EH4 2XU, UK; s1446279@sms.ed.ac.uk (J.L.); charlene.kay@ed.ac.uk (C.K.); mike.dixon@ed.ac.uk (J.M.D.); arran.turnbull@ed.ac.uk (A.K.T.); 2Translational Oncology Research Group, MRC Institute of Genetics and Cancer, Western General Hospital, University of Edinburgh, Edinburgh EH4 2XU, UK; james.meehan@ed.ac.uk (J.M.); mark.gray@ed.ac.uk (M.G.)

**Keywords:** breast cancer, predictive tools, prognostic tools, translational research, IL6ST, gp130, cytokine signalling

## Abstract

Novel biomarkers are needed to continue to improve breast cancer clinical management and outcome. IL6-like cytokines, whose pleiotropic functions include roles in many hallmarks of malignancy, rely on the signal transducer IL6ST (gp130) for all their signalling. To date, 10 separate independent studies based on the analysis of clinical breast cancer samples have identified IL6ST as a predictor. Consistent findings suggest that IL6ST is a positive prognostic factor and is associated with ER status. Interestingly, these studies include 4 multigene signatures (EndoPredict, EER4, IRSN-23 and 42GC) that incorporate IL6ST to predict risk of recurrence or outcome from endocrine or chemotherapy. Here we review the existing evidence on the promising predictive and prognostic value of IL6ST. We also discuss how this potential could be further translated into clinical practice beyond the EndoPredict tool, which is already available in the clinic. The most promising route to further exploit IL6ST’s promising predicting power will likely be through additional hybrid multifactor signatures that allow for more robust stratification of ER+ breast tumours into discrete groups with distinct outcomes, thus enabling greater refinement of the treatment-selection process.

## 1. Background: The Essential Role of Biomarkers in Breast Cancer

Breast cancer (BC) is a heterogeneous disease comprising well-characterised molecular subtypes that differ in their oncogenic drivers, pathogenesis and prognosis. Clinical management and outcome have improved considerably over time, in part due to the identification and clinical application of biological markers (or biomarkers), which have been defined as “characteristics that can be objectively measured and evaluated as indicators of certain normal biological processes, pathogenic processes, or pharmacologic responses to a therapeutic intervention” [1]. Biomarkers can be classified as prognostic, when they indicate the likelihood of an event such as disease recurrence or progression, or predictive, when they indicate the likelihood of response or resistance to a given treatment [2]. They can be clinical or histopathological factors, such as patient or tumour characteristics, or molecular markers, such as the expression level of a certain protein or gene or the presence or frequency of a genomic event (e.g., a mutation).

Molecular biomarkers are often molecules playing a role in processes such as disease progression or treatment response. Thus, they may act as surrogates for the activity of a given driver and provide insight into the complex underlying tumour biology. A biomarker might be utilised qualitatively or quantitatively, as a continuous variable or with discrete cut-offs, alone or in combination with other markers in the form of multifactor tests or signatures. In their different capacities, biomarkers are highly valuable in disease detection, staging, monitoring or prognosis estimation and they can guide the treatment selection and decision-making process in the management of many cancers, including breast [3].

The foremost examples of BC biomarkers are the oestrogen receptor α (ERα or ER) and the human epidermal growth factor receptor 2 (HER2), which indicate differences in prognosis and predict responsiveness to endocrine and HER2-targeted therapies, respectively. Assessment of ER and HER2 status has long been mandatory for all new BC diagnoses to help guide treatment selection [4] and has considerably improved prognosis and survival for patients with hormone-dependent and HER2-overexpressing BC. Importantly, the role of ER and HER2 as biomarkers continues to evolve, with growing evidence on different genomic aberrations contributing to the development of treatment resistance [5]. Research on ER mutations in particular has been extensive, with their prevalence and clinical implications being assessed in several retrospective and currently-ongoing prospective trials, making their translation into clinical practice in the near future a strong possibility [6,7].

Research over the last two decades has led to the identification of numerous other molecular biomarkers. These include proteins referred to as cancer antigens, such as CA15-3, CA19-9, CA27-29, the carcinoembryonic antigen (CEA) or mucin-like carcinoma antigen (MCA), which can be measured in patient serum to enable early detection and prognostic assessment [8,9,10,11]. In addition to well-established genomic markers such as BRCA1/2 mutations [12], translational studies continue to describe aberrations and rearrangements that could serve as prognostic factors or actionable targets [13]. Recent studies have also highlighted microRNAs as molecules with an emerging potential as biomarkers due to their complex regulatory role in breast cancer [14,15,16].

Many challenges still remain in the clinical management of BC. Evidence suggests that the current diagnostic tools and available biomarkers fail to sufficiently discriminate the underlying heterogeneity of the disease. Both basic and translational research continue to add to our understanding of the complex and evolving biology, shedding light on the pathways and mechanisms involved in phenomena such as the development of acquired resistance to treatment. Biomarker discovery studies can identify promising candidates with prognostic or predictive value which will be essential to continue to improve BC management and outcome. Here we will review the evidence on one molecule in particular, the interleukin-6 signal transducer (IL6ST), which has emerged as a novel and exciting BC biomarker in recent years.

## 2. The IL6-Like Cytokine Family and Its Signalling in Breast Cancer

Interleukin-6 (IL6) is the best characterised cytokine of a class that also includes interleukin-11 (IL11), interleukin-31 (IL31), ciliary neurotrophic factor (CNTF), leukemia inhibitory factor (LIF), oncostatin M (OSM), cardiotrophin 1 (CT1), cardiotrophin-like cytokine (CLC) and neuropoietin (NPN). This group of cytokines, with similar structural and functional features, are normally referred to as the IL6 or IL6-like family [17,18]. They are also known as the gp130 family, after the shared transmembrane signalling receptor glycoprotein 130, which acts as a signal transducer in all signalling by this cytokine family. Each oligomeric signalling complex includes one or more gp130 molecules, depending on the cytokine. This signal transducer is also known as CD130, IL-6 receptor subunit β (IL6Rβ) or IL6 signal transducer (IL6ST, which is also its gene name). For naming consistency, in this review we will refer to this cytokine group as the IL6-like family and to the signal transducer as IL6ST. 

Indeed, the common dependence on IL6ST for signalling is the defining characteristic of this cytokine family. The signal transducer is ubiquitously expressed in all cell types [19] and has been shown to be essential for survival in knockout in vivo studies in mice [20]. IL6-like cytokines act as extracellular ligands that bind the membrane-bound IL6ST and different non-signalling receptors with high affinity (see Figure 1 for diagram). This leads to the formation of signalling complexes including IL6ST homo- or heterodimers (depending on the cytokine). The cytoplasmic portions of the dimerised transducers then trigger intracellular signalling primarily through tyrosine kinases in the JAK family, such as JAK1 and JAK2, which are constitutively associated with IL6ST. JAK1/2 dimerisation and autophosphorylation lead to signalling through 3 major pathways: (i) the Janus-activated kinase – signal transducer and activator of transcription (JAK/STAT) pathway, (ii) the Ras-Raf mitogen-activated protein kinase (MAPK/MERK/ERK) signalling cascade, and (iii) the phosphoinositol-3 kinase – protein kinase B/Akt pathway (PI3K/AKT). 

Through the sophisticated signalling machinery downstream of IL6ST, subject to complex modulation by a wide range of regulatory mechanisms, interacting factors and cross-talking pathways, IL6-like cytokines are among the most pleiotropic protein families in the human body. They have been shown to play important roles in homeostasis, immunity, inflammation and disease pathogenesis, including a well-established role in numerous cancer types [21]. This includes breast neoplasms, where they are involved in many of the hallmarks of cancer development and progression. As the prototypical member of this cytokine family, the role of IL6 in particular has been extensively studied [22,23,24]: although in vitro studies have reported both pro- and anti-tumourigenic effects, the role of IL6 as a negative prognosticator in BC is firmly established [25,26], with circulating serum levels in patients correlating with disease stage and higher levels being associated with worse prognosis and survival and poorer response to chemo- and endocrine therapy [27,28,29,30]. Other IL6-like cytokines have been shown to play important roles in BC, including IL11 and LIF, which can promote migration and metastasis [22,23,31].

Although signalling through a shared transducer can entail some redundancy in the roles of different IL6-like cytokines, there is also extensive evidence of functional specificity for the different ligands in vivo: specific cytokines can exert unique functions, which can result from a balance of distinct, often contrasting effects; additionally, one cytokine might elicit different responses in different cell types [32,33,34]. This balance of redundancy and specificity is an inherent trait of this cytokine family and is likely made possible by differences in the expression patterns of different ligands and receptors across varying tissues and cell types [32,35].

The involvement of IL6-like cytokines in many BC-related processes has highlighted their promise not only as biomarkers, but also as therapeutic targets [36]; the signal transducer itself, its ligands, co-receptors or downstream interacting factors could be modulated using either novel agents or re-purposed similarly-targeted drugs already used in the clinic for the management of other pathologies. While some such agents are currently in pre-clinical or clinical testing for their use in BC [23,24,25,36,37], targeting of such a complex and pleiotropic signalling axis might prove difficult, as effective inhibition will need to be fine-tuned to achieve sufficient specificity. The central signalling role in BC also suggests potential for the identification of novel biomarkers, as already established for serum levels of IL6. As the central transducer of this family, IL6ST expression could be an indicator of overall signalling activity in this cytokine class and has been identified as a potential predictor in several biomarker discovery studies. The next sections will summarise the evidence to date on the role of IL6ST as a biomarker in BC, which has led to its incorporation into several molecular signatures with prognostic and predictive value.

## 3. IL6ST as an Independent Predictor in BC

To date, ten independent studies based on the analysis of clinical samples by different research groups have reported IL6ST as a predictor with potential clinical utility in BC (see Table 1). Six of these studies assessed the role of the signal transducer as an independent biomarker, showing an association between IL6ST expression and prognosis in BC.

In their study of primary breast carcinomas, Karczewska et al. found that 5-year rates of both overall (OS) and disease-free survival (DFS) were significantly higher in the IL6ST-positive (IL6ST+) compared to IL6ST-negative groups (90% vs. 9% and 88% vs. 0%, respectively) [38]. Similar trends were observed for IL6 and its non-signalling receptor α (IL6R), although the survival differences were more marked in relation to IL6ST expression. Indeed, univariate analysis found significant differences in OS and DFS associated with IL6ST status (*p* < 0.0001). Subgroup analysis showed IL6ST was independent from other well-established prognostic factors, while multivariate analysis found that IL6ST expression was the strongest positive prognostic factor. The researchers concluded that IL6ST expression was associated with earlier stages of BC but, in advanced stages, its active expression correlated with better prognosis. This study also showed that IL6ST expression was negatively correlated with both lymph node status and tumour size. These findings were consistent with a more recent study by Klahan et al., which found that IL6ST expression was significantly downregulated in breast tumours with lympho-vascular invasion (*p* = 0.037) [39].

In a study of triple-negative (negative status for ER, HER2 and progesterone receptor (PR)) BC (TNBC) across 3 independent sample cohorts, Mathe et al. found that IL6ST was one of only 4 genes that were differentially expressed between normal and BC tissues and which also differed in expression between TNBC and ER-positive (ER+) BC subtypes [40]. They showed that IL6ST expression was lower in TNBC than in the ER+ group, but also that higher IL6ST levels were significantly associated (*p* < 0.05) with better OS in TNBC patients. Subsequent validation on a larger cohort of publicly-available cases also showed that higher IL6ST expression was associated with significantly increased relapse-free survival [41].

In their assessment of cases from 2 large publicly-available BC datasets, Fertig et al. found that IL6ST was significantly overexpressed (*p* = 2 × 10 − 16) in tumours classified as luminal A or luminal B intrinsic subtypes (characterised by ER+/PR+ status) [42], consistent with previous reports of lower levels in TNBC. Survival analysis showed a trend towards longer survival in IL6ST-expressing luminal A tumours (*p* = 0.06) but not in other subtypes (Table 1).

**Table 1 jpm-11-00618-t001:** Summary of studies reporting on the role of IL6ST as a biomarker in breast cancer, including study cohorts and main findings. Studies are listed in chronological order of the original publication. See Table 2 for further description of the multifactor signatures. All the described associations achieved statistical significance (at least *p* < 0.05).

Original Publication	Study Type	Study Cohorts	Associations Reported	Main Predictive or Prognostic Value
Karczewska et al. (2000) [38]	Independent biomarker	75 PBCs who received surgery +/− adjuvant therapy.	IL6ST expression strongly correlates with earlier disease stages. In advanced stages, IL6ST expression is associated with better prognosis and higher OS and DFS rates. IL6ST negatively correlates with lymph node status and tumour size. IL6ST is independent from other well established clinicopathological factors.	IL6ST is a positive prognostic factor.
Tozlu et al. (2006) [43]	Independent biomarker	PBCs who received surgery (+ ET for ER+): - 12 in screening set. - 36 in validation set.	IL6ST is a perfect discriminator of ER+ status.	IL6ST is predictive for ER status and likely endocrine responsiveness.
Filipits et al. (2011) [44]	Molecular signatures: **EP** and **EPclin**	Original cohorts of ER+/HER2- BCs treated with ET: - 964 in training set. - 2948 in validation sets [44,45,46,47]. - ER+/HER2- BCs chemotherapy study [48]: - 2630 in ET alone arm. - 1116 in ET + chemotherapy arm.	EP and EPclin scores (linked to lower IL6ST expression) are continuous predictors of the risk of distant recurrence. EPclin is also prognostic for disease recurrence in patients who received chemotherapy, regardless of menopausal status. Patients with higher EPclin score derive benefit from the addition of chemotherapy to ET.	EP and EPclin stratify into risk groups that are prognostic for risk of distant recurrence at 5, 10 and 15 years in ER+/HER2- patients. EPclin is also prognostic for LRFS. EPclin high-risk group is predictive for chemotherapy benefit in pre- and postmenopausal ER+/HER2- patients.
Sota et al. (2014) [49]	Molecular signature: **IRSN-23**	PBCs who received NAC: - 58 in training set. - 59 in validation set. - 901 in external validation set (publicly-available data).	Higher IL6ST is associated with lack of pCR from NAC. IRSN-23 classifies into Gp-R and Gp-NR groups, with differential response to NAC.	IRSN-23 signature stratifies into groups predictive of response to NAC, regardless of BC subtype of chemotherapy regimen.
Andres et al. (2014) [50]	Independent biomarker	Tumour marker analysis: - 98 male BCs (publicly-available data). - 18,366 female BCs (publicly-available data). - Gene expression analysis validation: - 12 male BCs. - 233 female BCs.	IL6ST expression is significantly elevated in male BCs compared to female malignancies. IL6ST correlates with ER expression.	
Mathe et al. (2015) [40]	Independent biomarker	Screening set: - 33 TNBCs; 17/33 with matched normal tissue, 15/33 with lymph node metastases. - Validation sets: - 16 TNBCs; 4/16 with matched normal tissue - 255 non-TNBC. - Independent validation sets [41]: - 255 (publicly-available data) TNBCs. - 148 TNBCs.	IL6ST expression is associated with longer survival. IL6ST expression is lower in TNBC than ER+ tumours.	IL6ST is prognostic for OS and RFS in TNBC.
Fertig et al. (2015) [42]	Independent biomarker	638 + 897 PBCs from publicly-available sets.	IL6ST expression is higher in luminal tumours (ER+/PR+) than in other BC subtypes. Positive trend towards longer survival in IL6ST+ luminal A tumours.	
Turnbull et al. (2015) [51]	Molecular signatures: **EER4, EA2** and **EA2clin**	EER4 cohort of ER+ postmenopausal IBCs treated with NET & ET: - 73 training set. - 44 validation set. - EA/EA2clin study cohort of ER+ IBCs treated with NET & ET [52,53]: - 186 postmenopausal. - 51 premenopausal.	IL6ST alone is an independent predictor of response to AIs. EER4 predicts response to AIs with greater accuracy and also predict RFS and BCSS. EA2 and EA2clin predict outcome from adjuvant ET with greater accuracy and also predict RFS and BCSS. EA2 also predicts outcome in premenopausal women. EA2clin predicts treatment response regardless of ET regimen.	IL6ST is an independent predictive marker for AI response in ER+/HER2- patients. EER4 further improves on this predictive ability. Models are prognostic of outcome (RFS, BCSS) from adjuvant ET response, regardless of menopausal status or ET regimen in ER+/HER2- patients.
Klahan et al. (2017) [39]	Independent biomarker	108 pretreated IBCs: - 79 LVI+ - 29 LVI-	IL6ST correlates with LVI in samples without lymph node metastasis and perineural invasion.	
Tsunashima et al. (2018) [54]	Molecular signature: **42GC**	ER+ BCs treated with ET who recurred: - 177 training set (from publicly-available sets); 84 LR, 93 NLR. - 201 validation set; 137 LR, 84 NLR.	Higher IL6ST is associated with lower risk of early recurrence but higher risk of late recurrence. 42GC classified intro LR and NLR groups, with differential risk of recurrence over time. could predict late recurrence	42GC stratifies into prognostic groups for risk of early and late recurrence in ER+ BC intervals.

**42GC**, 42-gene classifier; **AI**, aromatase inhibitor; **BC**, breast cancer; **BCSS**, BC-specific survival; **DFS**, disease-free survival; **EA2**, EndoAdjuvant 2; **EA2clin**, EndoAdjuvant 2 clinical; **EER4**, Edinburgh EndoResponse 4; **EP**, EndoPredict; **EPclin**, EndoPredict clinical; **ER**, oestrogen receptor; **ET**, endocrine therapy; **Gp-NR**, genomically-predicted non-responders; **Gp-R**, genomically-predicted responders; **HER2**, human epidermal growth factor receptor 2; **IBC**: invasive BC; **IRSN-23**, immune-related 23-gene signature for NAC; **LN+**: lymph node positive status; **LR**, late recurrence-like; **LRFR**, local recurrence-free survival; **LVI**, lympho-vascular invasion; **NAC**, neoadjuvant chemotherapy; **NET**, neoadjuvant ET; **NLR**, non-late recurrence-like; **OS**, overall survival; **PBC**, primary BC; **pCR**, pathological complete response; **PR**, progesterone receptor; **RFS**, recurrence-free survival; **TNBC**, triple negative BC.

Consistent with previous observations, Tozlu et al. also showed that the expression of IL6ST and ER were significantly associated (*p* = 1.4 × 10^−6^) and positive expression of the signal transducer was highly predictive of ER+ status, perfectly discriminating between ER+ and ER- tumours (area under the receiver operating characteristic curve = 1) [43]. Andres et al. also reported that IL6ST expression was associated with ER+ status (*p* < 0.05), in addition to finding that it was upregulated in male breast tumours compared to those from female patients (*p* < 0.05) [50].

## 4. Molecular Signatures Incorporating IL6ST

The most relevant work in the literature comes from studies that developed molecular signatures including IL6ST and which showed prognostic or predictive power. This section describes the development to date of these signatures (also summarised in Table 1 and Table 2 and Figure 2 and Figure 3), likely to be the best avenues for the clinical application of IL6ST as a biomarker in BC.

### 4.1. EndoPredict and EPclin Scores for Prediction of Risk of Distant Recurrence

In 2011, Filipits et al. presented a prognostic signature named EndoPredict (EP) that predicted the likelihood of distant recurrence (DR) at 5 and 10-years in patients with ER+/HER2- BC treated with endocrine therapy (ET) alone [44]. This molecular classifier was based on the assessment of the expression level of 8 cancer-related genes (3 linked to proliferation and 5 linked to ER signalling, including IL6ST) and 3 reference genes using reverse transcription quantitative polymerase chain reaction (RT-qPCR). Higher IL6ST expression led to a lower EP score and, consequently, lower associated risk of recurrence and better prognosis. This molecular score was combined with lymph node status and tumour size to provide a hybrid score named EPclin (see Figure 2 for diagram).

Initial independent validation showed that the continuous EP score was an independent predictor of DR in multivariate analysis and also provided additional prognostic information. EPclin was able to stratify patients into low (score < 3.3) and high-risk (score ≥ 3.3) groups with significantly different 10-year rates of DR (4 vs. 22–28%, *p* < 0.001), outperforming conventional clinicopathological parameters [44]. Further analysis of one validation cohort also showed that the two risk groups exhibited statistically significant different rates of local recurrence-free survival (LRFS) at 10 years, but also concluded that EPclin was not useful to help tailor local therapy [55]. Another validation study showed that EP was also an independent prognostic parameter in both pre- and postmenopausal patients who received chemotherapy, although it could not predict differences in efficacy between drug regimens [45]. Another study showed that EP was significantly associated with distant metastasis, with higher expression of the module of genes linked to ER signalling in particular contributing to reduced risk [46]. The multigene classifier was subsequently revised, adding 1 control gene for a final 12-gene molecular assay [56]. Subsequent studies validated the performance characteristics and robustness of the test, supporting its reliability for decentralised molecular assessment of luminal breast tumours [56,57,58].

Further work assessed the potential of EP to predict benefit from the addition of adjuvant chemotherapy to ET in both pre- and postmenopausal ER+/HER2- BC patients [48]. EPclin was highly prognostic for 10-year DR in both patients who received ET alone and in those that received it in combination with chemotherapy (*p* < 0.0001 for both groups). Results also showed that 10-year DR risk was significantly lower among patients with a high EPclin score who received chemotherapy, but no differences were found between the treatment groups for patients with low EPclin score. This suggested that a high EPclin score can predict benefit from chemotherapy in ER+/HER2- BC patients and could be used to guide treatment selection.

A recent study reassessed the prognostic power of the assay in the original validation cohorts including longer clinical follow-up to assess distant recurrence-free rates at 10 and 15 years [47]. Results showed that the EPclin score also had significant prognostic value in predicting 15-year DR, irrespective of nodal status. Additionally, they suggested that this score could help guide treatment selection: a low EPclin score may help identify patients with reduced risk of recurrence who could safely forgo adjuvant chemotherapy at diagnosis (particularly any low-risk patients with nodal involvement who would be likely to receive chemotherapy without added benefit) or extended ET at the 5-year mark.

### 4.2. Immune-Related 23-Gene Signature for Prediction of Response to Neoadjuvant Chemotherapy

Sota et al. constructed a signature based on gene expression microarray analysis which included 23 probes (for 19 genes, with IL6ST being represented by 3 probes) to predict the response to neoadjuvant chemotherapy (NAC) in BC patients [49]. The immune-related 23-gene signature for NAC (IRSN-23) classified patients into 2 groups, the genomically-predicted responders (Gp-R) and non-responders (Gp-NR). The Gp-R group had significantly higher rates of pathological complete response (pCR) after NAC in both the internal (38 vs. 0%, *p* = 1.04 × 10^−6^) and external validation (40 vs. 11%, *p* = 4.98 × 10^−23^) sets. This study did not select patients based on ER status and the results showed that IRSN-23 held prognostic power regardless of the patients’ receptor status or chemotherapy regimen. Importantly, IL6ST was the most statistically significant marker of poorer response to NAC in the signature, with its higher expression being associated with non-pCR (*p* < 0.005). This is consistent with the previous study showing that patients with lower EP and EPclin scores (and, thus, higher IL6ST expression) derived no benefit from NAC [41].

### 4.3. Edinburgh EndoResponse4, EndoAdjuvant2 and EA2clin for Prediction of Response to and Outcome from Adjuvant Endocrine Therapy

Work from our group has led to the development of tools for the prediction of response to ET in postmenopausal ER+ BC patients who received neoadjuvant endocrine therapy (NET). The Edinburgh EndoResponse4 (EER4) predictive model is a 4-gene classifier incorporating the expression level of 2 genes (including IL6ST) before treatment and another 2 genes after 2 weeks of NET to classify patients into discrete responder (R) and non-responder (NR) groups [51] (see Figure 3 for diagram). IL6ST+ status alone could predict good clinical response to aromatase inhibitors (AI) with high accuracy (85%). This was further improved by EER4, which included IL6ST as its primary classifier, in both the training (96%) and independent validation (91%) sets. EER4 was also shown to significantly predict recurrence-free (RFS) (*p* = 0.029) and BC-specific survival (BCSS) (*p* = 0.009). We also showed that this 4-marker test could be performed using qPCR or immunohistochemistry (IHC).

Subsequent work has continued to revise this model. EndoAdjuvant2 (EA2) consisted of an improved tool incorporating IHC-based assessment of 2 markers at different timepoints: IL6ST at diagnosis and the proliferation-related MCM4 at 2 weeks on-treatment [52]. EA2 clinical (EA2clin) is a hybrid tool combining EA2 with clinical factors, namely node status, tumour size and grade, also included in the Nottingham Prognostic Index (NPI) tool [59] (see Figure 3 for diagram). Interestingly, EA2 (but not EA2clin) was shown to accurately predict outcome from adjuvant ET in both postmenopausal (*p* = 0.001 and *p* = 0.016 for RFS and BCSS, respectively) and premenopausal women (*p* = 0.002 and *p* = 0.016 for RFS and BCSS, respectively). EA2clin showed the best performance in postmenopausal patients, outperforming both EA2 and NPI and accurately predicting outcome (*p* < 0.001 for both RFS and BCSS) regardless of the type of adjuvant ET received [53].

### 4.4. 42-Gene Classifier for Prediction Risk of Late Recurrence

The team that developed IRSN-23 also sought to generate a molecular assay for prediction of recurrence in ER+ BC treated with ET alone [54]. Tsunashima et al. constructed a 42-gene classifier (42GC) including 42 probes (37 genes, including IL6ST represented by 5 probes) identified from gene expression microarray data. This signature was used to classify patients into the late-recurrence-like (LR) and non-late recurrence-like (NLR) groups. IL6ST was the most statistically significant marker in the 42GC signature (*p* < 0.005), with the LR group presenting higher expression of the signal transducer.

Results showed that the prognosis of the 2 groups identified was different and varied over time. The LR group showed significantly higher rates of late recurrence (5–15 years) and significantly lower rates of early recurrence (0–5 years) when compared to NLR in both the training (*p* = 0.006 and *p* = 1.6 × 10^−13^, respectively) and validation (*p* = 0.02 and *p* = 5.7 × 10^−5^, respectively) sets. Based on the previously established link between IL6ST expression and response to ET [51], the researchers hypothesised that the higher IL6ST expression in the LR group suggested these patients would benefit from extended ET (Table 2).

## 5. Discussion

In the search for novel candidate biomarkers to continue to improve the management and outcome of BC, IL6ST has emerged as a signal transducer with potential value as a predictor. We sought to review studies to date based on patient samples and data, rather than pre-clinical studies, in order to focus on results with greater clinical relevance and, thus, more likely translation into practice. We identified ten independent studies to date reporting IL6ST as a prognostic or predictive BC biomarker, either alone or as part of a multi-marker signature. Overall, these studies analysed samples and/or data from over 30,000 patients including both prospective processing of tissue samples (n > 9000) and analysis of publicly-available data (n > 25,000). Here we have reviewed and summarised this research, from which several trends have emerged.

Firstly, IL6ST seems to be a positive prognostic marker, with its higher expression being associated with better prognosis and survival rates both as an independent marker [38,40,41,42] and when the signal transducer is incorporated into a multi-factor signature [44,47,51]. The signal transducer has also been shown to be significantly associated with a number of other biomarkers. One prominent association reported in numerous studies across the literature is the correlation between IL6ST expression and ER+ status [40,42,43,50]. IL6ST levels have also been shown to negatively correlate with tumour size [38] and grade [51], as well as with nodal [38] or lymphovascular invasion [39].

The importance of this signal transducer in disease is well established, given its role as the signalling cornerstone for IL6-like cytokines, whose pleiotropic functions include regulation of cellular processes linked to the hallmarks of BC [22]. Interestingly, some authors had previously suggested IL6ST might instead correlate with malignancy, given its higher expression in infiltrating cancers compared with in situ or benign lesions [60]. This observation would also be in line with the fact that IL6, whose activity is dependent on IL6ST expression, has been shown to correlate with poorer prognosis in BC. Nevertheless, the complexity of the IL6ST signalling axis and the many cross-talking pathways modulating its downstream effects prevent a straight-forward description of the biological and clinical significance of this signal transducer. Indeed, the literature summarised here provides consistent evidence of IL6ST as a positive prognostic biomarker. This also includes through its association with other markers, which suggests IL6ST expression is linked to a lower risk of invasion, metastasis and recurrence and, thus, to better prognosis. IL6ST expression has also been shown to be higher in luminal tumours, which are characterised by a better clinical prognosis than other BC subtypes.

Multigene signatures including IL6ST have demonstrated prognostic value. Specifically, EP/EPclin and 42GC have shown that IL6ST expression is associated with differences in recurrence rates. The prognostic signature EPclin is already a well-established molecular assay, having been validated and reviewed with longer follow-up, and is currently commercially-available from Myriad Genetics. Expert panels in the USA and Europe have endorsed the use of EPclin to help guide treatment selection for patients with ER+/HER2-, node-negative early BC when the indication for adjuvant therapy is uncertain [61,62,63]. Most recently, the American National Comprehensive Cancer Network endorsed its use for prognostic purposes [64], while guidelines from the UK’s National Institute of Health and Care Excellence state that EPclin may be used for patients who had an intermediate risk of DR in other tools such as NPI [65]. The extent of the use of EPclin will vary between countries depending on each territory’s recommendations and health system. While it is relatively early to assess its adoption into practice in most countries, a recent prospective assessment estimated 63% cost-effectiveness for EP (versus usual care) within the Canadian health system [66], although it should be mentioned that this study also reported greater probability of cost-effectiveness for other clinically-available gene expression profiling tests. Some research has sought to assess the potential effect on the treatment decision-making process: two retrospective studies found that EPclin would lead to changes in therapy recommendation, either escalation or de-escalation, in ~35% of cases [67,68]; another study assessing how physicians’ level of experience affected the decision-making process found that EPclin could be particularly beneficial to help less experienced physicians prevent over or undertreatment [69].

Interestingly, EPclin and 42GC research reported some contrasting findings. Evidence from EPclin studies was consistent with previous research [70,71,72,73,74] in showing that recurrence risk trends (low vs. high-risk) were consistent across time; as it pertains to IL6ST, patients with higher expression (i.e., lower score) showed decreased rates of both early and later distant recurrences. In contrast, 42GC results suggested the risk of recurrence might change overtime; thus, higher IL6ST expression was associated with the LR group of patients, with lower risk of early recurrence but higher risk of later recurrence. This could be interpreted as being in line with the described correlation between IL6ST and ER, as ER+ BC has been shown to sustain risk of recurrence over a longer period of time post-treatment than other subtypes [75].

The 42GC study used a distinct approach that likely contributed to these diverging findings, as the team specifically focused on the biological differences between malignancies that lead to early and late recurrences in their study design and supervised analysis. This differentiation in 42GC would mean a more complex prognostic role and patient stratification, compared with the other recurrence-predicting tool EPclin, in which IL6ST was very clearly a positive prognostic marker whose higher expression was linked to lower rates of distant recurrence and better prognosis [44]. Interestingly, researchers also drew different conclusions from their findings. EPclin researchers interpreted their evidence as indicative that patients in the high-IL6ST/low-risk group may be able to safely forgo extended ET [47], while the 42GC researchers hypothesised that the higher IL6ST expression in the LR group suggested that these patients would benefit from extended ET, based on the previously established link between IL6ST expression and response to ET [51]. Despite these diverging evidence and conclusions, EPclin benefits from extensive validation and its already-established clinical use.

Other molecular signatures incorporating IL6ST have also been shown to hold predictive power, with potential to help guide the selection of endocrine and chemotherapy. While the pretreatment level of IL6ST alone was shown to be a good predictor of response to AIs, this predictive ability was further improved in the EER4 model, which incorporates IL6ST as its main classifier [51]. The revised tools EA2 and EA2clin have shown great accuracy and robustness in prediction of outcomes from treatment with adjuvant ET across several validation cohorts and, importantly, regardless of menopausal status and type of ET. These tools are also advantageous in that, unlike other molecular tests such as EP, they are based on IHC assessment and, thus, could be easily implemented in local laboratories. They also enable discrete risk stratification, making its interpretation for potential clinical application more straight-forward than continuous scores.

Evidence has also shown that IL6ST expression is predictive of a lack of response to chemotherapy. In the IRSN-23 signature, designed specifically to predict response to NAC, higher IL6ST expression was linked to a lack of pCR. In line with this, a recent study showed that ER+ BC patients with higher IL6ST expression and, consequently, lower EPclin scores did not benefit from the addition of chemotherapy to ET. While these results diverge from previous evidence in that IL6ST acts as a negative predictor, they are consistent with the fact that different BC subtypes, as well as some subsets within the ER+ BC population, will respond differently to chemotherapy. Indeed, while the IRSN-23 study did not select patients according to hormone receptor status, results showed that the Gp-NR group was significantly enriched for luminal breast tumours (*p* < 0.005), which would typically show less response to chemotherapy [76,77].

Finally, IL6ST might hold particular promise as a biomarker in ER+ disease. The link between IL6-like cytokines and oestrogen-related signalling in BC is already well documented [26,37] and, as summarised here, numerous studies have reported a correlation between both biomarkers. Evidence suggests that, in addition to its prognostic role, IL6ST might be a robust surrogate marker of active oestrogen signalling and, consequently, responsiveness to ET. Indeed, we have shown that IL6ST can identify subsets of breast lesions with active ER-dependent signalling within larger ER+ populations [78,79,80]. This suggests that the predictive value of IL6ST might partly emerge from the biomarker’s ability to discriminate the complex underlying biology of hormone-dependent disease, possibly enabling a finer stratification than histological assessment of ER status alone currently allows. In this way, IL6ST might serve as a marker to identify those ER+ tumours that are more likely to respond to readily-available endocrine therapy.

## 6. Conclusions

In recent years, IL6ST has emerged as a biomarker with prognostic and predictive value in BC. Although the complex role of IL6ST signalling in the disease might prevent the description of a simple mechanism behind this predictive value, there is extensive evidence that expression of this signal transducer is a positive prognostic factor in BC.

While current research efforts are investigating the potential of targeting the IL6ST signalling axis as a therapeutic approach, studies to date support the notion that the best route for exploiting IL6ST as biomarker in the clinical setting will be as part of multifactor hybrid signature and likely within the ER+ subset of the disease.

In this way, tools incorporating IL6ST could enable patient stratification into discrete groups that more accurately reflect the underlying biology of the disease and, consequently, better predict prognosis and the likelihood of treatment response. As with any tools of this type, successful clinical translation will necessitate prospective studies to both corroborate the prognostic and predictive ability of IL6ST and its related signatures, and to help define any potential clinical guidelines, particularly on whether lower-risk patients might be able to safely forgo neoadjuvant or extended therapy. Overall, with sufficient validation, tools incorporating IL6ST as a molecular biomarker could improve the management of BC by helping to make a better, more targeted use of the therapeutic strategies already available in the clinic.

## Figures and Tables

**Figure 1 jpm-11-00618-f001:**
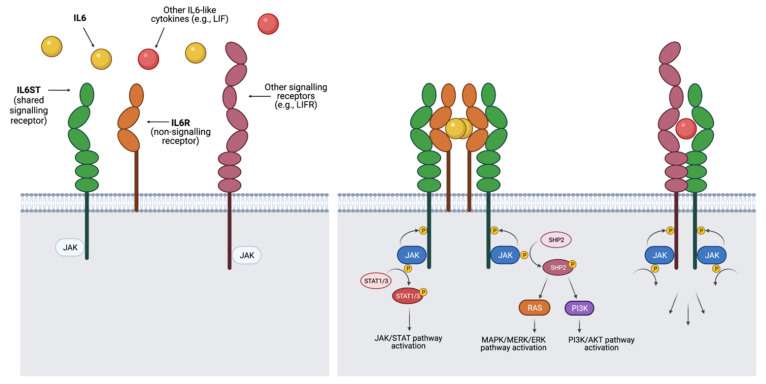
Summary of signalling by cytokines in the IL6-like family. Cytokines bind membrane-bound receptors with similar modular structures to form signalling complexes including 2 signalling receptors, of which at least 1 is always the shared signal transductor IL6ST. Dimerisation of these receptors leads to the activation of tyrosine kinases bound to their cytoplasmic sections, which in turn trigger a signalling cascade that can activate 3 pathways with known roles in breast cancer: JAK/STAT, MAPK/MERK/ERK and PI3K/AKT. Signalling complexes are different for each cytokine in the family. For example, IL6 is recruited by the non-signalling (lacking cytoplasmic domains) receptor IL6 receptor α (IL6R), leading to the formation of a hexameric signalling complex including an IL6ST homodimer. Other cytokines in the family form complexes comprising heterodimers (with IL6ST and a different signalling receptor) with or without the need of a non-signalling receptor, depending on the cytokine. Members of the IL6-like family can exert specific functions due to variations in ligand and receptor concentrations and in the activity of modulating signals across different cell and tissue types.

**Figure 2 jpm-11-00618-f002:**
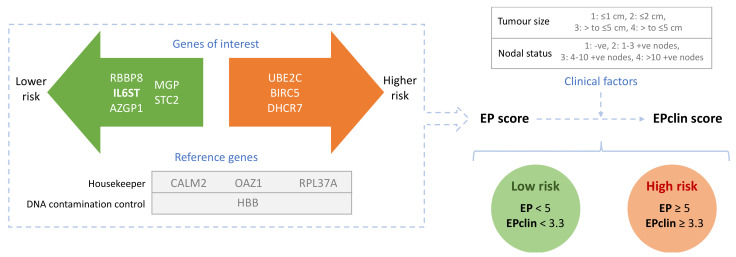
Summary of the markers included in the EndoPredict (EP) molecular signature and the EPclin hybrid signature, which combines EP with clinical factors. Both continuous scores allow for stratification into discrete risk groups with differential rates of distance recurrence.

**Figure 3 jpm-11-00618-f003:**
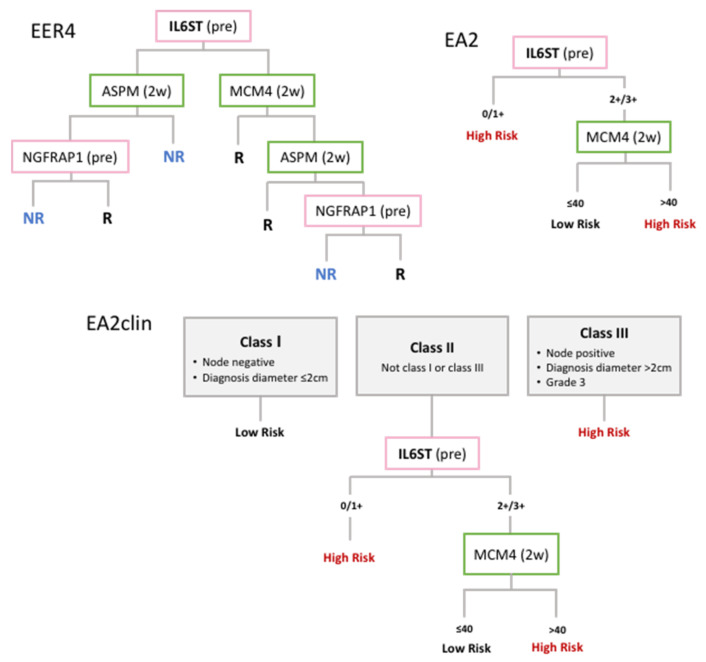
Summary of the markers included in the different predictive models developed in Edinburgh: the 4-gene classifier Edinburgh EndoResponse 4 (EER4) incorporates the expression level of 2 genes at pretreament (pre) and 2 genes after 2 weeks of neoadjuvant endocrine therapy (2w); this was simplified into the EndoAdjuvant2 (EA2) signature, which uses IHC assessment of the 2 main classifiers to stratify cases into discrete risk groups; EndoAdjuvant2 clinical (EA2clin) combined EA2 with clinical factors to produce a more accurate hybrid model.

**Table 2 jpm-11-00618-t002:** Summary of molecular signatures incorporating IL6ST. The markers included in each model are listed, as well as its prognostic or predictive value. See Figure 2 and Figure 3 for further description of the hybrid multifactor signatures.

Original Publication	Signature	Biomarkers Incorporated in the Signature		Clinical Significance
Filipits et al. (2011) [44]	**EndoPredict**	Low risk-associated (surrogates for ER signalling/cell differentiation): RBBP8, **IL6ST**, AZGP1, MGP, STC2		- Stratifies into prognostic groups for risk of distant recurrence in ER+/HER2- BC patients - Predictive for benefit from the addition of chemotherapy in the high-risk group in ER+/HER2- patients
		High risk-associated (surrogates for proliferation/cell cycle): BIRC5, UBE2C, DHCR7		
		Housekeeper genes: CALM2, OAZ1, RPL37A		
		Control gene: HBB		
	**EPclin**	Clinical factors: Lymph node status, tumour size		
		Molecular factors: EndoPredict genes		
		LR-associated:**IL6ST** (5 probes), NPY1R, ELOVL5, ASAH1 (2 probes), ALDH6A1, SYBU, RAB5C, PTP4A2, HSPA2, SLC7A8 ADRA2A, MYCBP, CX3CR1, ERCC1, DNAJA3, NINJ1, C4orf43, IFI35, ZNF688, SNX1, CREBL2, HPN, NME3, PDHB, NKX3-1, DEXI, GSTM3, LCMT1		
Sota et al. (2014) [49]	**IRSN-23**	Non-pCR-associated: **IL6ST** (3 probes), CX3CR1, ZEB1 (2 probes), SEMA3C, HFE, EDA		- Stratifies into groups predictive for response to NAC.
		pCR-associated: CARD9, IDO1, CXCL9, PNP, CXCL11 (2 probes), CEBPB, CD83, CD1D, CTSC, CXCL10, IGHG1, VEGFA, CR2		
Turnbull et al. (2016) [51]	**EER4**	Pretreatment levels: **IL6ST**, NGFRAP1		- Predictive for response to AIs in postmenopausal ER+/HER2- BC patients. - Prognostic for long term outcome (RFS and BCSS) in postmenopausal ER+/HER2- BC patients treated with AIs.
		2-week levels: ASPM, MCM4		
	**EA2**	Pretreatment levels: **IL6ST**		- Prognostic for long term outcome (RFS and BCSS) in ER+/HER2- BC patients treated with ET, regardless of menopausal status.
		2-week levels: MCM4		
	**EA2clin**	Clinical factors	Lymph node involvement, tumour size and tumour grade	- Prognostic for long term outcome (RFS and BCSS) in ER+/HER2- BC patients treated with ET, regardless of ET regimen.
		Molecular factors	Pretreament level: **IL6ST** 2-week level: MCM4	
Tsunashima et al. (2018) [54]	**42GC**	NLR-associated: KLF7, STS, RALA, SMURF2, OXTR, ABCC10, ASAP2, CALB2, OPA1		- Stratifies into prognostic groups for risk of early and late recurrence in ER+ BC.
		LR-associated: **IL6ST** (5 probes), NPY1R, ELOVL5, ASAH1 (2 probes), ALDH6A1, SYBU, RAB5C, PTP4A2, HSPA2, SLC7A8 ADRA2A, MYCBP, CX3CR1, ERCC1, DNAJA3, NINJ1, C4orf43, IFI35, ZNF688, SNX1, CREBL2, HPN, NME3, PDHB, NKX3-1, DEXI, GSTM3, LCMT1		

**42GC**, 42-gene classifier; **AI**, aromatase inhibitor; **BC**, breast cancer; **BCSS**, BC-specific survival; **EA2**, EndoAdjuvant 2; **EA2clin**, EndoAdjuvant 2 clinical; **EER4**, Edinburgh EndoResponse 4; **ER**, oestrogen receptor; **HER2**, human epidermal growth factor receptor 2; **EA2**, EndoAdjuvant2; **EA2clin**, EndoAdjuvant2 clinical; **ET**, endocrine therapy; **IRSN-23**, immune-related 23-gene signature for NAC; **NAC**, neoadjuvant chemotherapy; **pCR**, pathological complete response; **RFS**, recurrence-free survival.

## Data Availability

Not applicable.

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
