# Peer review of "The Signal Transducer IL6ST (gp130) as a Predictive and Prognostic Biomarker in Breast Cancer"

_jpm, 2021, doi:10.3390/jpm11070618_

Round 1

Reviewer 1 Report

Martínez-Pérez and colleagues proposed an interesting review article on the diagnostic and prognostic role of IL6ST in breast cancer. The authors clearly described the physiological role of IL6ST describing also its involvement in the pathogenesis of breast cancer. In addition, they carefully examined all the existing studies evaluating the prognostic value of IL6ST in different molecular subtypes of breast cancer. Overall, the manuscript is well written and the review well-structured. Some parts need supporting references. Below are reported some minor comments that will improve the quality of the manuscript:
1) In the following paragraphs, the authors should indicate some examples of clinical, protein or molecular biomarkers: “They can be clinical or histopathological factors, such as patient or tumour characteristics, or molecular markers, such as the expression level of a certain protein or gene or the presence or frequency of a genomic event (e.g., a mutation). Molecular biomarkers are often molecules playing a role in processes such as disease progression or treatment response. Thus, they may act as surrogates for the activity of a given driver and provide insight into the complex underlying tumour biology. A biomarker might be utilised qualitatively or quantitatively, as a continuous variable or with discrete cut-offs, alone or in combination with other markers in the form of multifactor tests or signatures. In their different capacities, biomarkers are highly valuable in disease detection, staging, monitoring or prognosis estimation and they can guide the treatment selection and decision-making process in the management of many cancers, including breast[3].”. Please, briefly mention the diagnostic and prognostic role of protein biomarkers (CEA, Ck19, MUC-1, CA15-3, CA27.29 etc.) and molecular biomarkers (mainly BRCA mutations and microRNAs). For this purpose, please see the following and other studies:
- PMID: 28081538
- PMID: 32911851
- PMID: 26987529
2) Please provide references for the entire paragraph: “Interleukin-6 (IL6) is the best characterised cytokine of a class that also includes interleukin-11 (IL11), interleukin-31 (IL31), ciliary neurotrophic factor (CNTF), leukemia inhibitory factor (LIF), oncostatin M (OSM), cardiotrophin 1 (CT1), cardiotrophin-like cytokine (CLC) and neuropoietin (NPN). This group of cytokines, with similar structural and functional features, are normally referred to as the IL6 or IL6-like family. They are also known as the gp130 family, after the shared transmembrane signalling receptor glycoprotein 130, which acts as a signal transducer in all signalling by this cytokine family. Each oligomeric signalling complex includes one or more gp130 molecules, depending on the cytokine. This signal transducer is also known as CD130, IL-6 receptor subunit β (IL6Rβ) or IL6 signal transducer (IL6ST, which is also its gene name). For naming consistency, in this review we will refer to this cytokine group as the IL6-like family and to the signal transducer as IL6ST.”;
3) In table 2 please add also information about the predictive value of these signatures and the statistical relevance of the analyses of these genes;
4) The authors fully elucidated the role of IL6ST in breast cancer as well as its diagnostic and prognostic potential. However, they should provide also some information about the potential role of IL6 (functionally associated with IL6ST) and indicate if gene polymorphism may influence IL6ST or IL6 functions in breast cancer;
5) In the Discussion section, please avoid redundancy with the data presented in the previous paragraphs (lines 417-440).

Author Response

We thank the reviewer for their helpful suggestions. Below we detail the changes made to the revised manuscript:

1. Please, briefly mention the diagnostic and prognostic role of protein biomarkers (CEA, Ck19, MUC-1, CA15-3, CA27.29 etc.) and molecular biomarkers (mainly BRCA mutations and microRNAs).

A paragraph has been added to section 1 providing some more information on the biomarkers suggested, as well as additional references.

2. Please provide references for the entire paragraph...

We have added references to a couple of publications that provide greater detail on the IL6-like cytokine family for which this paragraph provides an overview.

3. In table 2 please add also information about the predictive value of these signatures and the statistical relevance of the analyses of these genes;

Table 2 provides further details (namely, marker lists) on the multifactor signatures listed in table 1. As the predictive/prognostic value of each signature is already listed in the first table, including the information again would be redundant. We do not delve into the details of statistical relevance in table 2 because, as stated in table 1, all reported studies were showed to be statistically significant and also specific P-values have been provided within the text where relevant.

4. The authors fully elucidated the role of IL6ST in breast cancer as well as its diagnostic and prognostic potential. However, they should provide also some information about the potential role of IL6 (functionally associated with IL6ST) and indicate if gene polymorphism may influence IL6ST or IL6 functions in breast cancer.

While we agree the topics suggested are interesting, we intended to keep this review quite focused on IL6ST and IL6ST-including multifactor signatures based on clinical samples only. We have included some references to other recent reviews, but we believe that the discussion of cytokines or other related receptors might deviate too much from our intended scope. There is some evidence in the literature about IL6ST polymorphisms, but this is largely in pre-clinical studies and also has not been linked to the predictive/prognostic role of the biomarker that we aimed to focus on. 

5. In the Discussion section, please avoid redundancy with the data presented in the previous paragraphs (lines 417-440).

While it is true that this section repeats some of the information provided before, this is because we are now discussing the findings from different studies against each other. We things this is useful for the scope of the review and enables us to discuss some important nuances such as differences in how easily each tool could be implemented.

Reviewer 2 Report

In this study the authors have tried to described the prognostic and predictive role of IL6ST. The article is well written and documented and they were able to synthetized most of the work done related to this biomarker. Still, there are some minor points that need to be corrected:

  1. Please write the numbers 6 and 10 with letter in the section 3.
  2. Please verify the abbreviations in Table 1 you have EP EndoPredict and EP= EndoPredict clinical
  3. In table 1 you have "groups prognostic for"  please rephrase in "prognostic groups for". The same in Table 3 
  4. AT the end of Discussion section you have the following phrases: "Authors should discuss the results and how they are interpreted form ....." I think these phrases do not belong here and should be removed.
  5. Please rephrase the Conclusion section to include your conclusion related to everything you described and not  parts of discussions and no references should be added here.
  6. Please describe the exact contribution of all the others authors or remove them if they had no contribution to this article.  

Author Response

We thank the reviewer for their helpful suggestions. Below we detail the changes made to the revised manuscript:

1. Please write the numbers 6 and 10 with letter in the section 3.

This has been amended.

2. Please verify the abbreviations in Table 1 you have EP EndoPredict and EP= EndoPredict clinical.

Thanks for bringing up this typo. This has been corrected.

3. In table 1 you have "groups prognostic for"  please rephrase in "prognostic groups for". The same in Table 3 

This has been amended in both tables.

4. AT the end of Discussion section you have the following phrases: "Authors should discuss the results and how they are interpreted form ....." I think these phrases do not belong here and should be removed.

Thanks for bringing up this error (this sentence was part of the instructions in the JPM template used). This has been removed.

5. Please rephrase the Conclusion section to include your conclusion related to everything you described and not parts of discussions and no references should be added here.

We have re-written the conclusion to avoid as much redundancy with the discussion. Instead, we highlight what we think are the future prospects and main take-home messages of the review. References have been removed from this section.

6. Please describe the exact contribution of all the others authors or remove them if they had no contribution to this article.  

Contributions have been expanded.